# *Clerodendranthus spicatus* (Thunb.) Water Extracts Reduce Lipid Accumulation and Oxidative Stress in the *Caenorhabditis elegans*

**DOI:** 10.3390/ijms25179655

**Published:** 2024-09-06

**Authors:** Xian Xiao, Fanhua Wu, Bing Wang, Zeping Cai, Lanying Wang, Yunfei Zhang, Xudong Yu, Yanping Luo

**Affiliations:** 1School of Tropical Agriculture and Forestry, Hainan University, Haikou 570228, China; ynxiaoxian@163.com (X.X.); wang0307bing@163.com (B.W.); 992995@hainanu.edu.cn (Z.C.); daivemuwly@126.com (L.W.); diyfzhang@163.com (Y.Z.); 2School of Life Sciences, Hainan University, Haikou 570228, China; wrwfh@163.com

**Keywords:** *Caenorhabditis elegans*, *Clerodendranthus spicatus*, kidney tea, anti-ageing, antioxidant, resistance

## Abstract

*Clerodendranthus spicatus* (Thunb.) (Kidney tea) is a very distinctive ethnic herbal medicine in China. Its leaves are widely used as a healthy tea. Many previous studies have demonstrated its various longevity-promoting effects; however, the safety and specific health-promoting effects of *Clerodendranthus spicatus* (*C. spicatus*) as a dietary supplement remain unclear. In order to understand the effect of *C. spicatus* on the longevity of *Caenorhabditis elegans* (*C. elegans*), we evaluated its role in *C. elegans*; *C. spicatus* water extracts (CSw) were analyzed for the major components and the effects on *C. elegans* were investigated from physiological and biochemical to molecular levels; CSw contain significant phenolic components (primarily rosmarinic acid and eugenolinic acid) and flavonoids (primarily quercetin and isorhamnetin) and can increase the lifespan of *C. elegans*. Further investigations showed that CSw modulate stress resistance and lipid metabolism through influencing DAF-16/FoxO (DAF-16), Heat shock factor 1 (HSF-1), and Nuclear Hormone Receptor-49 (NHR-49) signalling pathways; CSw can improve the antioxidant and hypolipidemic activity of *C. elegans* and prolong the lifespan of *C. elegans* (with the best effect at low concentrations). Therefore, the recommended daily use of *C. spicatus* should be considered when consuming it as a healthy tea on a daily basis.

## 1. Introduction

Aging encompasses an inevitable neurodegenerative process accompanied by a gradual deterioration of physiological functions that can result in a number of diseases, including hypertension, cancer, and metabolic syndrome [1]. In a previous investigation, an analysis of data from multiple databases showed that metabolic syndrome was associated with a 2-fold increased risk of developing cardiovascular disease and stroke, and a 1.5-fold increased risk of all-cause mortality [2,3]. In today’s society, the creation of anti-ageing products has taken on significant importance. An increasing amount of research indicates that including some natural botanical ingredients in the diet may be a useful way to delay aging and enhance health [4]. The use of botanical dietary supplements with fewer side effects than pharmaceutical interventions has sparked interest in drug candidates with anti-ageing and anti-ageing-related disease properties [5]. However, in recent years, health problems caused by botanical dietary supplements have increased. For example, the improper use of He-Shou-Wu [*Reynoutria multiflora* (Thunb.) Moldenke] and Huang-Yao-Zi (*Dioscorea bulbifera* L.) have caused liver damage [6]. Therefore, more attention should be paid to the health-promoting effects of herbs as dietary supplements and whether there are any adverse effects.

*Clerodendranthus spicatus* (Thunb.) (*C*. *spicatus*), also named “maoxucao” in China, is a plant growing in tropical and subtropical regions, and while less-studied abroad, some Southeast Asian countries use it to treat diabetes, urinary tract diseases, and inflammation of the kidney and it is a very popular healthy tea in China [7]; corresponding products have been marketed both domestically and internationally [8,9,10]. *C*. *spicatus* is also known as *Orthosiphon aristatus* and *Orthosiphon stamineuso* and refers to “kidney tea” in the literature and pharmacopoeia [11]. Studies have demonstrated many of its life-prolonging properties. *C*. *spicatus* has a significant protective effect on 6-OHDA (6-hydroxydopamine) in a rat model of Parkinson’s disease and in a cellular model, and its mechanism of action is related to the reduction in oxidative-stress-induced cellular damage [12]. In addition, *C*. *spicatus* has a significant inhibitory effect on the growth of renal cancer cells, and its main mode of action may be through the regulation of the cell cycle and apoptosis, so as to achieve a growth inhibition effect [13]. In a study of *C*. *spicatus* reducing appetite and reducing fat accumulation, *C*. *spicatus* was found to reduce food intake and visceral fat mass in rats [14]. In addition, many studies have shown that *C*. *spicatus* has good antioxidant activity and the active ingredients mainly include 3′-hydroxy-5,6,7,4′-tetramethoxyflavone, rosemarinic acid, apigenin, and caffeic acid. *C*. *spicatus* water, ethanol, and methanol extracts have shown good free radical scavenging antioxidant activity in vitro [15,16,17]. Despite the availability of several *C*. *spicatus*-related dietary supplement products on the global market, there are few studies on the antioxidant activity in vivo or health-promoting effects of long-term *C*. *spicatus* consumption.

For research on aging, *C. elegans* is the preferred model. It is very easy to raise and the food is mainly *E. coli* OP50. Its life cycle is very fast, taking 12 h to form an embryo, 60 h to develop into an adult, and its lifespan is about 21 days. Moreover, it is possible to obtain plenty of nematodes that are all the same age, which enhances the reproducibility of studies involving vital analysis. The body is translucent, facilitating the observation of various tissues in the body. For example, the eggs, pharynx, gut, muscle cells, and embryonic cells of nematode worms are easily observed. Additionally, it also can be used to observe fluorescent markers [18]. Insulin/IGF-1 (IIS) is the first definitive longevity pathway identified in *C. elegans*, and three transcription factors downstream of the IIS pathway—DAF-16/FoxO, Nrf-2/SKN-1, and Heat shock factor 1 (HSF-1)—regulate the expression of a series of lifespan-related factors, stress-related factors, metabolism-related factors, and protein homeostasis factors, which are ideal targets for screening of anti-ageing substances [19,20,21]. When IIS signalling is inhibited, DAF-16 and SKN-1 can generate nuclear translocation and activate the expression of downstream target genes, and affect the life span and response to various kinds of stress in *C. elegans* [22]. Many studies have shown that *C. spicatus* has good antioxidant activity; however, the role of *C. spicatus* in modulating oxidative damage and prolonging lifespan in *C. elegans* is not known. Therefore, in this study, we focused on the modulation of antioxidant pathways and the relationship between the IIS pathway and antioxidant activity-inducing signalling pathways to explain the effect of *C. spicatus* on lifespan.

## 2. Results

### 2.1. Effects of CSw on the Health Life Span of C. elegans

One of the most logical and significant biological indicators for evaluating the aging process is life span. We treated *C. elegans* with 0, 20, 50, 100, 500, and 1000 μg/mL CSw, from the beginning of the L4 stage until day 8, and then transferred worms to common NGM until death under normal growth conditions. The mean life span of *C. elegans* with 20 μg/mL increased to 21.46 ± 0.71 days (maximum life span: 29 days), with 50 μg/mL it increased to 20.63 ± 0.77 days (maximum life span: 28 days), with 100 μg/mL it increased to 19.40 ± 0.73 days (maximum life span: 26 days), with 500 μg/mL it increased to 18.46 ± 0.62 days (maximum life span: 24 days), and with 1000 μg/mL it increased to 18.33 ± 0.88 days (maximum life span: 25 days), which expanded the life span of *C. elegans* compared with the 0 μg/mL CSw-treated group (17.00 ± 0.89 days; maximum life span: 23 days; Figure 1A, Appendix A). These findings suggested that CSw can extend the lifespan of *C. elegans*, with the 20 and 50 μg/mL CSw-treated groups showing the greatest extension. A widely recognised indicator of aging is the accumulation of lipofuscin, an oxidation byproduct of lysosomal degradation. On days 4 and 8, the lipofuscin concentration was measured. The fluorescence intensity of the CSw-treated group was significantly less than that of the control group, suggesting that CSw have the potential to delay senescence (Figure 1B,C).

Fertility (egg-laying capacity) not only reflects the reproductive toxicity of a compound in *C. elegans* but also reflects whether the compound promotes extended life span by inhibiting reproduction. Compared with the 0 μg/mL CSw-treated group, other concentrations of CSw did not inhibit the egg production of *C. elegans* and showed a facilitative effect. In the 500 μg/mL CSw-treated group, egg production was increased by 166%, indicating that CSw extended the life span without sacrificing reproduction (Figure 2A).

An excellent anti-ageing product should not only extend the lifespan but also keep the body healthy. We used the motor ability and pharyngeal pumping rate to evaluate the effects of CSw on the health parameters of *C. elegans*, and the head swing and body bending frequency is the most commonly used detection index of *C. elegans* motor ability. The rate of pharyngeal pumping can reflect the feeding status of nematodes. Both young and old nematodes’ pharyngeal pump rates were increased by the addition of CSw (Figure 2B). As illustrated in Figure 2C,D, the 100 μg/mL CSw significantly increased the body length and reduced the body width of *C. elegans*, and 100 μg/mL CSw significantly enhanced the movement of both young and old nematodes (Figure 2E,F). As shown in the results above, CSw enhanced nematode health indicators.

### 2.2. Effects of CSw on Stress Resistance in C. elegans

A decline in stress tolerance is a common side effect of aging, and the ability to withstand stress in response to particular environmental stimuli can reflect the physiological state of the body. This ability is frequently used to describe health status in old age. To investigate whether CSw improve stress resistance, we determined the survival rates of *C. elegans* under heat stress (37 °C), 250 μM paraquinone-induced oxidative stress, and UV stress. The ability of nematodes to withstand heat stress was significantly increased by CSw when compared to the 0 μg/mL CSw-treated group, and the group that was given 500 μg/mL of CSw saw a survival rate increase of 25.83% over the 0 μg/mL CSw-treated group (Figure 3A). The ability of nematodes to withstand oxidative stress was significantly increased by CSw when compared to the 0 μg/mL CSw-treated group, and the group that was given 500 μg/mL of CSw saw a survival rate increase of 65.00% over the 0 μg/mL CSw-treated group (Figure 3B). The ability of nematodes to withstand UV stress was significantly increased by CSw when compared to the 0 μg/mL CSw-treated group, and the group that was given 500 μg/mL of CSw saw a survival rate increase of 14.28% over the 0 μg/mL CSw-treated group (Figure 3C). As shown in the results above, CSw enhance stress resistance in *C. elegans*.

To further explore the reason why CSw enhance stress resistance in *C. elegans*, we determined the reactive oxygen species (ROS) level. Treatment of nematodes for 7 days with CSw significantly lowered ROS level in vivo (Figure 3D,E). Malondialdehyde (MDA), a measure of lipid peroxidation, was significantly reduced in nematodes after CSw treatment (Figure 3E). The antioxidant defence system is mainly responsible for removing ROS. Therefore, we determined the activities of two important antioxidant enzymes: Superoxide dismutase (SOD) and Catalase (CAT). The levels of SOD and CAT activity significantly increased after the addition of CSw, partly explaining the decrease in ROS in *C. elegans* (Figure 3F). The above results indicate that CSw activate the antioxidant system in *C. elegans*.

### 2.3. Effects of CSw on the Fat Content in C. elegans

A high concentration of CSw resulted in an increase in body length and decrease in body width. We evaluated the effects of CSw on the overall lipid storage. Nematodes were cultured as described in Section 4.8.1, and the high-cholesterol medium was prepared as described in Section 4.1. The photographic results were consistent with the quantitative results, which showed a significant decrease in nematode fat deposition after 7 days of treatment with 100 and 500 μg/mL CSw (Figure 4A,B). Triglycerides (TG) are important components of the nematode body fat and are widely present in the intestinal and subcutaneous tissues in the form of lipid droplets. The TG content of nematodes was significantly reduced after 7 days of CSw treatment (Figure 4C). To further investigate the impacts of CSw on the lipid metabolism of *C. elegans*, we examined the content of free fatty acids (FFA), which are the products of fat hydrolysis and substrates for fat synthesis. FFA content was significantly reduced in nematodes after 7 days of treatment with CSw (Figure 4D). The above results indicated that CSw have a beneficial effect on fat reduction in *C. elegans*.

Considering that the 20 μg/mL CSw-treated group had the best life-prolongation effect on *C. elegans* and showed certain lipid-lowering effects in the previous fat deposition experiment, this concentration was selected for the subsequent experiment. A functional homolog of mammalian peroxisome proliferator-activated receptors that control fatty acid oxidation, fatty acid desaturation, and fatty acid binding and transport is Nuclear Hormone Receptor-49 (NHR-49) in *C. elegans*. The mRNA expression of *nhr-49* was significantly up-regulated 1.55-fold in the 20 μg/mL CSw group compared with the model (+cholesterol) group. *acs-2* and *ech-1* are both important genes that control mitochondrial oxidation. The results of the experiment showed that 20 μg/mL of CSw up-regulated *acs-2* expression 2.63-fold but not *ech-1* (Figure 5D).

The upstream gene of *acs-2* is *nhr-49*. Further research is required to determine whether CSw influence lipid metabolism via NHR-49. As a result, we looked into how CSw affected the expression of *acs-2* in the *nhr-49 (nr2041)* mutant. As anticipated, the expression of *acs-2* in the *nhr-49(nr2041)* mutant was unaffected by the CSw (Figure 5E). Meanwhile, GFP-labelled acs-2 worms (*acs-2::GFP*) were used to verify the results. The expression of *acs-2::GFP* after CSw treatment was not significantly different from that in the control, confirming that CSw can affect fat accumulation by up-regulating the expression of *acs-2* (Figure 5F,G).

### 2.4. Effects of CSw on Life-Span-Extension Signalling Pathway in C. elegans

The insulin pathway (IIS) is a classical and conserved senescence regulatory signalling pathway that regulates the activities of three transcription factors—FoxO/DAF-16, HSF-1, and Nrf-2/SKN-1—thereby inducing a series of stress resistance, homeostatic regulation, and metabolism genes. To investigate the role of the IIS pathway in the life-span-extending effect of CSw, we analysed the effect of CSw on the mRNA levels of the three transcription factors through qRT-PCR. The results indicated that 20 μg/mL CSw up-regulated *daf-16* by 3.67-fold and *hsf-1* by 2.85-fold. No significant change in mRNA levels of *skn-1* was found. Given that mRNA levels cannot accurately reflect transcription factor activity, the transcription levels of *daf-16* downstream force-resistance-related genes *sod-3*, *ctl-1*, *ctl-2*, and *ctl-3* as well as *hsf-1* downstream gene *hsp-16.2* and the *skn-1* classical target genes *gcs-1* and *gst-4* were subsequently quantified. The results indicated that 20 μg/mL of CSw up-regulated *sod-3* (3.52-fold), *ctl-1* (7.73-fold), *ctl-2* (6.03-fold), *ctl-3* (1.58-fold), and *hsp-16.2* (2.54-fold) in *C. elegans*, with no significant changes in *gcs-1* and *gst-4* transcript levels (Figure 6A–C).

When transcription factors are activated, they translocate from the cytoplasm to the nucleus, thereby binding to the DNA promoter binding region and inducing the expression of target genes. We investigated the effect of CSw on the nuclear translocation of transcription factors by using nematodes with the fluorescent tags of transcription factors. The results showed that the CSw treatment significantly induced DAF-16 transcription factor nuclear translocation (TJ356 nematode) but no significant effect on SKN-1 (LD1 nematode) transcription factor translocation was observed (Figure 6D,E).

Gene-deficient mutants were selected for life-span analysis to further investigate the role of transcription factors in the CSw-induced life-span-extension effect. The results showed that CSw treatment failed to extend the life span of the *daf-16* mutant (CF1038 nematode) and *hsf-1* mutant (PS3551 nematode) compared with the control group, indicating that the life-span-extension effect of CSw was dependent on DAF-16 and HSF-1 transcription factors. In contrast, the CSw-treated group significantly prolonged the life span of the *skn-1* mutant (EU1 nematode), indicating that the life-extending effect of CSw was independent of the SKN-1 transcription factor. Meanwhile, the lifespan of the oxidation-sensitive nematode mev-1 (TK22 nematode) was prolonged (Figure 7A–D). In addition, we measured the endogenous ROS levels in nematodes after administration and showed that CSw administration decreased ROS levels in skn-1 mutant (EU1 nematodes) and had no effect on the ROS levels in the daf-16 mutant (CF1038 nematodes) and hsf-1 mutant (PS3551 nematodes) (Figure 7E). To further verify that DAF-16 and HSF-1 transcription factors play an important role in the anti-ageing activity of CSw, we detected the expression of the fluorescent marker protein in the transgenic nematode *sod-3::GFP* and *hsp-16.2::GFP.* The positive puncta of *sod-3::GFP* and *hsp-16.2::GFP* nematodes in the intestine were counted. We found that CSw significantly increased the number of bright spots in the intestine (Figure 7F,G), further indicating the KTw life-prolonging effect is dependent on DAF- 16/FOXO and HSF-1 but not SKN-1.

### 2.5. Identification of CSw Main Compounds

For the components in CSw, positive and negative chromatograms were obtained, where a total of 43 molecules were identified (Appendix A). Chemical structures were mainly composed of 12 phenolic compounds (e.g., rosmarinic and syringic acid), 11 flavonoids (e.g., kaempferide and naringenin), 12 terpenoids (e.g., betaine and chicoric acid), 4 isopentenolipids, 2 fatty acids, and 2 coumarin derivatives. The Appendix A shows the tentative identification of the compounds together with the molecular formula and the *m/z* [M-H]^−^ or [M-H]^+^. We observed the presence of syringic, rosmarinic acid, and caryophyllene oxide as major compounds, followed by betaine and oleanolic acid (Table 1). 

## 3. Discussion

The aging process is due to the deterioration of a range of physiological functions associated with resistance to metabolism-related, diet-induced, and environmental stresses, which in turn increases the risk of developing age-related diseases [23]. There is growing evidence that natural bioactive substances, with antioxidant activity, may be effective in extending life span [24]. Thus, antioxidant properties and anti-ageing activity are often considered to be closely related. In traditional medicine, medicinal plants are mostly used as compound preparations to treat diseases and exert medical value, ignoring the fact that medicinal plants also serve as edible natural plants, which are an important food source in daily human life. Previous studies have reported that aqueous and ethanolic extracts of *C*. *spicatus* have a good in vitro antioxidant activity and scavenging effect on DPPH and ABTS free radicals [25]; however, the in vivo antioxidant activity of *C*. *spicatus* is unknown. *C*. *elegans* is a model organism widely used in senescence studies, and plant extracts were used to study their effects on the growth and senescence of *C*. *elegans* and the possible mechanism of action by adding them to the NGM medium in nematodes. Considering that *C*. *spicatus* as a daily drinking tea is mostly brewed and consumed in hot water, *C*. *spicatus water* extracts (CSw) were chosen for this study. The mean and maximum lifespan of naturally aging nematodes increased after CSw treatment. In addition, the motility and fertility data indicated that the concentrations of CSw used in this study were safe and did not lead to side effects.

In the longevity assay experiment, the treatment groups with the best life-extension effect were 20 μg/mL and 50 μg/mL, whereas in the lipofuscin and lipid assay experiments, the treatment group with the best effect was 500 μg/mL; a similar phenomenon has appeared in studies of ginseng polysaccharides, *Lycium barbarum*, and *astragalus* polysaccharides [24,26,27]. The reason for this difference may be that at low concentrations, CSw act as an antioxidant to prolong the lifespan of nematodes mainly by reducing oxidative stress. At high concentrations, CSw as an antioxidant may affect multiple cell signalling pathways or biological processes, not just oxidative stress. For example, it may modulate cellular metabolic activities, improve the intracellular environment, or regulate gene expression. These effects may lead to a range of physiological alterations including, but not limited to, enhanced immune function, improved apoptosis or cell cycle regulation, among other health parameters. This also suggests that CSw produce different biological effects at different doses, and the promotion of CSw as a daily drinking tea requires more safety evaluations that should be carried out in other mammalian experiments, such as in mice, and standards should be set for the recommended doses.

When the IIS signalling pathway is inhibited, DAF-16 can produce nuclear translocation and activate the expression of downstream target genes, and FOXO/DAF-16 and HSF-1 can act synergistically to promote lifespan extension [28,29]. In our study, the expression levels of *daf-16*, *hsf-1*, and their downstream genes were significantly up-regulated after CSw treatment (Figure 6A–C), and nuclear translocation of daf-16 was also promoted (Figure 6D,E). The up-regulation of gene expression levels was further verified by the results of visual detection with GFP-tagged mutants *sod-3::GFP* and *hsp-16.2::GFP* (Figure 7F,G). In addition, the lifespan of the *daf-16* and *hsf-1* gene-deletion mutants was not significantly different from that of the control (Figure 7A,B), suggesting that the longevity effect of CSw requires the synergistic effect of DAF-16 and HSF-1. A recent study showed that collagens are required for IIS signalling in mutants’ long lifespan and that overexpression of specific collagens extends wild-type lifespan [30,31].Therefore, the involvement of other factors in the CSw-mediated longevity effect cannot be excluded.

We found that body length increased and body width decreased in the CSw-treated *C*. *elegans*, suggesting that CSw may be involved in lipid metabolism in *C*. *elegans*. We determined the changes in lipid content of *C. elegans* after treatment with a combination of 5 mM cholesterol + 20 μg/mL CSw, and the results showed that CSw treatment reduced the lipid content of elegans (Figure 4A–D). Considering the effects of CSw on lipid metabolism in *C. elegans*, we further investigated the underlying mechanisms. Key pathways related to lipid metabolism include the NHR-49-mediated nuclear hormone signalling pathway; TGF-β and insulin metabolism pathway; SBP-1/MDT-15-mediated metabolism pathway; and TOR and hexosamine metabolism signalling pathway [21]. In our study, *nhr-49* and *acs-2* gene expression levels were up-regulated after CSw treatment, but not *ech-1* expression (Figure 5D), and the results of visual detection with the GFP-tagged mutant *acs-2::GFP* were consistent with the up-regulation of gene expression levels (Figure 5F,G). In the TGF-β and insulin metabolic pathways, CSw significantly down-regulated the expression of the downstream gene *daf-16* of the insulin signalling pathway in *C. elegans* but had no effect on the upstream genes *daf-2* and *age-1*, suggesting that the regulation of lipid metabolism by CSw may not require the insulin signalling pathway (Figure 5A). In this study, CSw significantly reduced the expression of *fat-5*, *fat-6,* and *fat-7*, suggesting that the lipid metabolism pathway controlled by SBP-1 and MDT-15 also contributes to CSw’s influence on lipid metabolism (Figure 5B). CSw had no effect on the TOR and hexosamine signalling pathways (Figure 5C). Meanwhile, we found that *daf-16* overexpression promotes the extension of lifespan in *C. elegans* and also causes fat accumulation. This suggests that fat accumulation and oxidative stress appear to interact.

The active constituents of *C*. *spicatus* are complex and currently, more than 100 chemical components such as flavonoids, phenolic acids, coumarins, lignans, terpenes, chromenes, steroids, glycosides, etc., have been isolated from *C*. *spicatus* [32]. In this study, 43 compounds were identified in CSw, and the compounds with higher content were rosemarinic acid, butyric acid, stigmasterol, chicoric acid, ursolic acid, oleanolic acid, caffeic acid, tangerine peel, sweet orange peel, ferulic acid, quercetin, and isorhamnetin. The health-promoting effects of many of these components have been demonstrated. Rosemarinic acid (RA) promotes longevity and locomotor activity in *C. elegans* and reduces fat storage in a dose-dependent manner without affecting fertility, and RA-induced longevity is associated with the genes *sod-3*, *sod-5*, *ctl-1*, *daf-16*, *ins-18*, *skn-1*, and *sek-1* [33]. Caffeic acid in combination with zinc sulphate synergistically exerts antioxidant effects and modulates glucose uptake and utilisation in L-6 myotubes and rat muscle tissue [34]. Caffeic acid restores the antioxidant capacity of trophoblast cells by enhancing GSH and efficiently scavenging ROS, attenuating oxidative DNA damage after hydrogen peroxide exposure, and reducing cytotoxicity, protein, and lipid peroxidation [35]. *C*. *elegans* supplemented with fermented onion extract and quercetin showed enhanced vital activity and reduced lipids and triglycerides [36]. Isoquercitrin has a wide range of pharmacological effects including cardiovascular protection, anti-inflammatory, anti-tumour, antioxidant, antibacterial, and antiviral activities [37]. Therefore, the abundance of flavonoid compounds and phenolic secondary metabolites in CSw may be the key active components in extending the lifespan of nematodes, but other active components may also have synergistic effects, and further studies are needed to investigate the health-promoting effects of the potential active components.

## 4. Materials and Methods

### 4.1. Materials and Strains

The test material *C*. *spicatus* was provided by Professor Xudong Yu of the School of Tropical Agriculture and Forestry, Hainan University, and the chromosome karyotype analysis of *C*. *spicatus* was completed in the previous period [38]. *C*. *spicatus* fresh leaves picked from Hainan University Danzhou base were made into *C*. *spicatus* according to four processes: spreading, killing, kneading, and drying. We pre-boiled approximately 2000 mL of deionized water, then accurately weighed 200 g of *C*. *spicatus* at a 10:1 ratio, soaked it in hot water for 10 min, and filtered it through several layers of gauze. Then, we added 2000 mL of hot water again for 15 min, combined the filtrates, and concentrated it under reduced pressure to obtain CSw. It was stored in the refrigerator at 4 °C until ready to use.

*C. elegans* were cultured using the Brenner method at 20 °C [39]. Wild-type N2 and uracil-deficient *E. coli* OP50 were donated by Rutgers University, New Jersey, USA. All mutant strains were obtained from the Caenorhabditis Genetic Center (CGC), including daf-16 (mu86), skn-1 (zu67), hsf-1 (sy441), mev-1 (kn1), nhr-49 (nr2041), *sod-3::GFP*, *hsp-16.2p::GFP*, *skn-1::GFP*, *daf-16::GFP*, and *acs-2::GFP*. All information about the mutant strains can be found on the GGC website. CSw were added to nematode growth medium (NGM) at 20, 50, 100, 500, and 1000 μg/mL. *C. elegans* were age-synchronized as described [40]. L1 stages of the same age were obtained. When not specifically stated, treatment with CSw was initiated when *C. elegans* were at stage L4, and the worms were treated for 7 days.

Preparation of the CSw medium: combined with the NGM preparation method, 5 mg/mL of CSw reserve solution was prepared, dissolved in sterile water, refrigerated for use, diluted into the required concentration solution when needed, and the amount of other substances added remained unchanged.

Preparation of the high-cholesterol medium (+cholesterol): combined with the NGM preparation method, the amount of cholesterol was changed to 1.25 mL (the amount of cholesterol in 250 mL ordinary NGM was 0.25 mL), and the amount of other substances added remained unchanged.

### 4.2. Life Span Assay

The life span assay was performed as described previously [41]. Synchronous L4-stage worms were moved to an NGM plate containing CSw (0, 20, 50, 100, 500, and 1000 μg/mL) until day 8; then, the CSw-treated worms were moved to standard NGM, survival rates were recorded daily until 100% mortality was reached, and the worms were transferred to freshly prepared plates daily to avoid reproduction. In each group, at least 30 nematodes were examined. Each experiment was conducted three times.

### 4.3. Analysis of Lipofuscin Accumulation

The amount of lipofuscin was measured as described earlier [40]. Synchronous L4-stage worms were moved to an NGM plate containing CSw (0, 20, 50, 100, and 500 µg/mL) until day 8 and then the nematodes were picked onto slides containing 2% agarose sheets, anaesthetised with 10 µL of levamisole hydrochloride solution at a concentration of 10 mmol/L, and pressed. The fluorescence images were observed under an inverted fluorescence microscope (excitation wavelength 351 nm; emission wavelength 420 nm). ImageJ 1.51 software was used to calculate the fluorescence intensity. In each group, at least 30 nematodes were examined. Each experiment was conducted three times.

### 4.4. Fertility Measurement

The number of offspring was determined as described previously [42]. L4-stage nematodes were transferred onto NGM plates containing CSw. Three nematodes were placed on one plate. The total number of eggs and the average number of eggs were recorded daily. On day 6, fertility was no longer measured. In each group, at least 30 nematodes were examined. Each experiment was conducted three times. 

### 4.5. Measurement of Health Parameters

#### 4.5.1. Pharyngeal Pumping Rate

Pharyngeal pumping rate was determined as previously described [43]. On days 4 and 8, measurements were taken, and the worms were moved to a new plate with OP50. Under a stereomicroscope (NIKON, Tokyo, Japan), each worm was observed for 30 s and pharyngeal motions were recorded. In each group, at least 30 nematodes were examined. Each experiment was conducted three times. 

#### 4.5.2. Head Swing and Body Bending Frequency

The measurements of head swing and body bending frequency were determined as described previously [43]. On days 4 and 8, measurements were taken, and each worm in good condition was randomly selected and moved to a new plate without OP50, followed by the addition of 60 μL of K buffer solution to allow the *C. elegans* to recover for 60 s. Each worm was observed for 20 s using a stereomicroscope. The movement of the head from one side to the other and back was defined as a head swing. The movement of *C. elegans* relative to one wavelength along the longitudinal axis of the body was defined as a body bend. In each group, at least 30 nematodes were examined. Each experiment was conducted three times.

#### 4.5.3. Body Length and Width

On day 8, the worms were washed with M9 buffer and moved to a 1.5 mL centrifuge tube where they were left for 3 min. The supernatant was then discarded. The procedures were repeated three times. The centrifuge tube containing the nematodes was placed in a water bath at a constant temperature of 55 °C for 3 min. The nematodes were placed on 2% molten agarose and then covered with glass slides, and the length and width of each worm were measured using ImageJ (NIH, Bethesda, MD, USA). Each group had a minimum of 30 worms counted. Each experiment was conducted three times.

### 4.6. Stress Resistance In Vivo

Stress resistance assays were carried out as described previously [44]. Synchronous L4-stage worms were moved to an NGM plate containing CSw (0, 20, 50, 100, and 500 μg/mL) until day 8; then, the worms were moved to new common NGM plates before being cultured at 37 °C for the heat stress assay. Survival rates were recorded hourly until 100% mortality was reached. For the antioxidant stress assay, the pretreated worms were transferred to a plate containing 250 μM pecan quinone. Survival rates were recorded hourly until 100% mortality was reached. The UV irradiation assay was conducted using 254 nm UV light (Huaqiang, Nanjing, China) as a stress inducer. Survival rates were recorded daily until 100% mortality was reached. Each group had a minimum of 30 worms counted. Each experiment was conducted three times.

### 4.7. Antioxidant Activity of CSw

Using a previously reported method, reactive oxygen species (ROS) levels in *C. elegans* were measured using 1 μM H2-DCF-DA (dihydrodichlorofluorescein diacetate, Solarbio, Beijing, China) [40]. CSw-treated worms were moved to a 24-well microtiter plate, followed by the addition of 1 mL of the H2-DCF-DA working solution, which was stained at 20 °C for 120 min in the dark. The stained worms were washed with M9 buffer, the supernatant was discarded, and 20 μL of 5 mmol/L levamisole hydrochloride was used to paralyze *C. elegans*. The paralyzed nematodes were then placed on 2% molten agarose and covered with glass slides. Using a microscope (excitation wavelength, 488 nm; emission wavelength, 510 nm), fluorescence images were captured. Enzyme activity and the content of SOD, CAT, and MDA in *C. elegans* were determined using kits (Solarbio, Beijing, China). A BCA kit (Solarbio, Beijing, China) was used to determine the protein concentration. Each experiment was conducted three times.

### 4.8. Fat Content Determination

#### 4.8.1. Oil-Red-O Staining

For obesity modelling, synchronous L4-stage worms were transferred to an NGM plate containing high cholesterol (model: +cholesterol). Some worms were grown on a standard NGM plate (with CSw 0 μg/mL), other worms were moved to an NGM plate (+cholesterol with CSw 20, 50, 100 and 500 μg/mL). The worms were treated with CSw for 7 days before being removed from the NGM plate and stained using a previously described method [45]. The stained worms were photographed using a fluorescence-inverted microscope and measured using ImageJ. At least 30 worms were measured in each group.

#### 4.8.2. Triglyceride Assay

*C. elegans* was treated as described in Section 4.8.1. *C. elegans* (0.1 g) to be tested were washed into an EP tube with M9 buffer on the 8th day. The triglyceride quantification kit (Solarbio, Beijing, China) was used to determine the triglyceride content by adding 1 mL of Reagent 1 into an EP tube, ice bath homogenization, 8000 g, centrifugation at 4 °C for 10 min, and then taking the supernatant solution for determination. Each experiment was conducted three times. Triglyceride content was normalized by determining the protein concentration using the BCA kit (Solarbio, Beijing, China).

#### 4.8.3. Free Fatty Acid Assay

*C. elegans* was treated as described in Section 4.8.1. *C. elegans* (0.1 g) to be tested were washed into an EP tube with M9 buffer on the 8th day. The free fatty acid kit (Solarbio, Beijing, China) was used to determine the triglyceride content by adding 1 mL of reagent into an EP tube, ice bath homogenization, 8000 g, centrifugation at 4 °C for 10 min, and then taking the supernatant solution for determination. Each experiment was conducted three times. Free fatty acid content was normalized by determining the protein concentration using the BCA kit (Solarbio, Beijing, China).

### 4.9. Quantitative Real-Time PCR

Total RNA was isolated from about approximately 0.2 mL of worms with a commercial kit (Tiangen, Beijing, China). After the RNA concentration was determined by NanoDrop, RNA samples were reverse transcribed to cDNA with a FastKing RT kit (Tiangen, Beijing, China). Quantitative real-time PCR (RT-qPCR) was conducted with 2× Q3 SYBR qPCR Master Mix (Tolobio, Nanjing, China). Real-time fluorescence quantitative PCR was used to detect changes in gene transcription levels of *daf-16*, *skn-1*, *hsf-1*, *nhr-49*, *fat-7*, *acs-2*, *ech-1*, *sod-3*, ctl-1, ctl-2, ctl-3, hsp-16.2, gcs-1, gst-4, daf-2, age-1, daf-15, ogt-1, oga-1, *sbp-1*, *mdt-15*, *fat-5*, and *fat-6* genes, and *gpd-1* was used as the internal reference gene (Appendix A shows the prime sequence). Each gene’s relative expression level was determined using the 2^−△△ct^ method. All of the experiments were analysed in triplicate.

### 4.10. Nuclear Translocation Assay

Nuclear translocation assay was performed as described [23,46]. At the L4 stage, the worms TJ356 (*daf-16::GFP*) and LD1 (*skn-1::GFP*) were transferred to NGM culture plates which included 20 μg/mL CSw and standard NGM as a control. After 7 days of incubation at 20 °C, the nematodes were picked onto agar pads and anaesthetised with 10 µL of levamisole hydrochloride solution at a concentration of 10 mmol/L. With a microscope, the nematodes *skn-1::GFP* and *daf-16::GFP* showed changes in their fluorescence. GFP expression patterns and defined as three types: “cytosolic”, “intermediate”, and “nuclear”. The distribution rate was calculated in three separate experiments with at least 30 individuals each time. 

### 4.11. hsp-16.2::GFP, sod-3::GFP, and acs2::GFP Expression Analysis

At the L4 stage, the worms TJ375 (*hsp-16.2*::GFP), CF1553 (*sod-3*::GFP), and WBM170 (*acs-2*::GFP) were transferred to NGM culture plates, which included 20 μg/mL CSw and standard NGM as a control. Transfer was carried out after incubation at 20 °C for 7 days, and anaesthesia was applied as described in Section 4.3. Green fluorescent aggregation was observed using a microscope (NIKON, Tokyo, Japan). At least 30 worms were measured in each group.

### 4.12. CSw Component Analysis

#### 4.12.1. Preparation of Test Solution

With a few modifications, the test solution was prepared as described previously [47]. We accurately weighed an appropriate amount of KTw into a 2 mL centrifuge tube; added 600 µL of CH_3_OH (stored at −20 °C) containing 2-Amino-3- (2-chloro-phenyl) -propionic acid (4 mg/L); vortexed it for 30 s; added a 100 mg glass bead; and placed it in a tissue grinder for 90 s at 60 Hz. It then underwent room-temperature ultrasound for 15 min; was centrifuged for 10 min at 12,000 rpm and 4 °C; and the supernatant was filtered by a 0.22 μm membrane and transferred into the detection bottle for LC-MS detection.

#### 4.12.2. Testing of On-Board

With a few modifications, the liquid chromatography and mass spectrometric conditions were carried out as previously described [48]. The column was maintained at 40 °C. The flow rate and injection volume were set at 0.25 mL/min and 2 μL, respectively. For the LC-ESI (+)-MS analysis, the mobile phases consisted of (B2) 0.1% formic acid in acetonitrile (*v*/*v*) and (A2) 0.1% formic acid in water (*v*/*v*). Separation was conducted under the following gradient: 0~1 min, 2% B2; 1~9 min, 2%~50% B2; 9~12 min, 50%~98% B2; 12~13.5 min, 98% B2; 13.5~14 min, 98%~2% B2; 14~20 min, 2% B2. For the LC-ESI (-)-MS analysis, the analytes were carried out with (B3) acetonitrile and (A3) ammonium formate (5mM). Separation was conducted under the following gradient: 0~1 min, 2% B3; 1~9 min, 2%~50% B3; 9~12 min, 50%~98% B3; 12~13.5 min, 98% B3; 13.5~14 min, 98%~2% B3; 14~17 min, 2% B3.

The mass spectrometry conditions were parameterised as follows: sheath gas pressure, 30 arb; aux gas flow, 10 arb; spray voltage, 3.50 kV and −2.50 kV for ESI(+) and ESI(-), respectively; capillary temperature, 325 °C; MS1 range, *m*/*z* 100–1000; MS1 resolving power, 60,000 FWHM; number of data-dependent scans per cycle, 4; MS/MS resolving power, 15,000 FWHM; normalized collision energy, 30%; dynamic exclusion time, automatic.

### 4.13. Statistical Analysis

The log-rank (Mantel–Cox) test was used to determine statistical significance for the life span analysis. The Kaplan–Meier method was used to create the lifespan graphs. The statistical significance was evaluated using one-way analysis of variance (ANOVA) followed by Duncan’s method for multiple comparisons with SPSS 22.0. Different letters (a, b, c, d) represent significant (*p* < 0.05) differences among different groups; the same letters indicate no significant differences between the two groups. Graphs were created using GraphPad Prism 8 and ImageJ 1.51 was used to analyze the fluorescence intensities.

## 5. Conclusions

We utilized *C. elegans* as a model to show the anti-ageing effects of CSw and investigate the underlying molecular mechanism. CSw significantly increased the mean and maximum life span of nematodes and improved the majority of their physiological aging markers. CSw regulate stress resistance and lipid metabolism in nematodes by regulating FOXO/DAF-16 and HSF-1 signalling pathways as well as the NHR-49 nuclear hormone signalling pathway, which ultimately promotes the life span of nematodes. In addition, CSw are rich in flavonoids and phenolics, which may contribute to their effects, and when drinking *C. spcates* daily, the dosage should be carefully controlled until a conclusion about their safety is reached. This study provides a theoretical foundation for the application and promotion of CS as a healthy drinking tea.

## Figures and Tables

**Figure 1 ijms-25-09655-f001:**
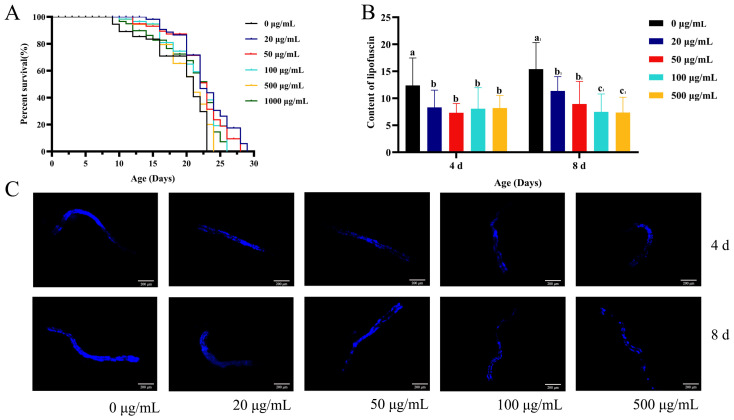
Effect of CSw with different concentrations on the life span of *C. elegans*. (**A**) Survival curves of *C. elegans* treated with different concentrations of CSw (0, 20, 50, 100, 500, and 1000 μg/mL); (**B**) qualitative observation of fluorescence intensity of lipofuscin; (**C**) representative pictures of lipofuscin. Different letters (a, b) represent significant (*p* < 0.05) differences among different groups; the same letters indicate no significant differences between the two groups. The same subscript number (a_1_, b_1_, c_1_) indicates that the significant differences analysis is performed within the same group.

**Figure 2 ijms-25-09655-f002:**
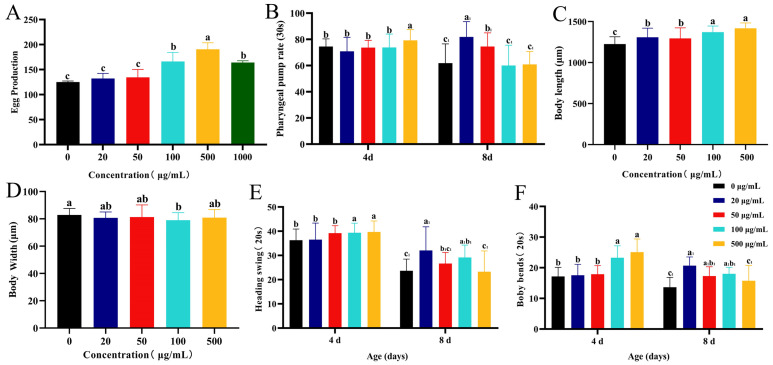
Effect of CSw with different concentrations on the health parameters of *C. elegans*. (**A**) Spawning; (**B**) pharyngeal pumping rate was counted for 30 s; (**C**) body length; (**D**) body width; (**E**) head swing; and (**F**) body bends counted under a dissecting microscope for 20 s. Different letters (a, b, c) represent significant (*p* < 0.05) differences among different groups; the same letters indicate no significant differences between the two groups. The same subscript number (a_1_, b_1_, c_1_) indicates that the significant differences analysis is performed within the same group.

**Figure 3 ijms-25-09655-f003:**
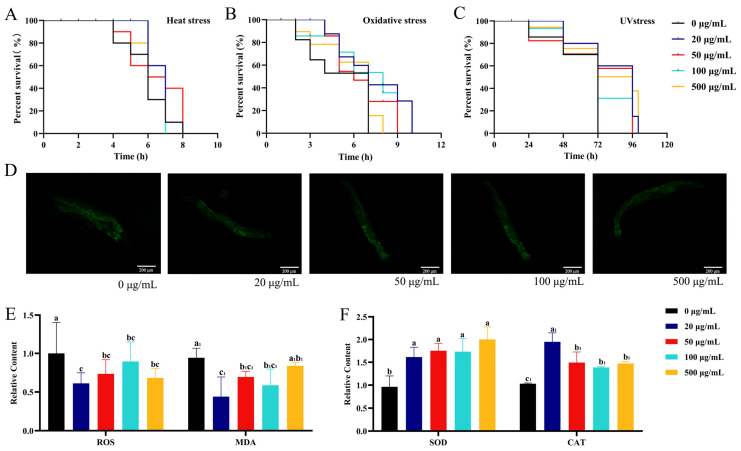
CSw activated the antioxidant system of *C. elegans*. Survival rates of *C. elegans* under (**A**) heat stress; (**B**) oxidative stress; and (**C**) UV stress; (**D**) representative picture of reactive oxygen species (ROS) in *C. elegans*; (**E**) ROS and Malondialdehyde (MDA) content; (**F**) Superoxide dismutase (SOD) and Catalase (CAT) activity. Different letters (a, b, c) represent significant (*p* < 0.05) differences among different groups; the same letters indicate no significant differences between the two groups. The same subscript number (a_1_, b_1_, c_1_) indicates that the significant differences analysis is performed within the same group.

**Figure 4 ijms-25-09655-f004:**
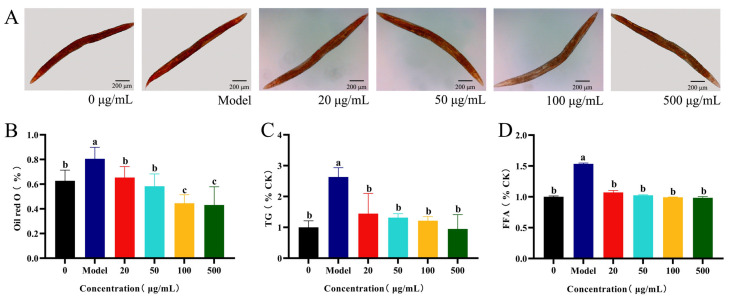
CSw reduce the fat content of *C. elegans*. (**A**) Photographs of *C. elegans* stained with oil red O; (**B**) quantitative results of oil red O; (**C**) triglyceride (TG) content; and (**D**) free fatty acid (FFA) content in *C. elegans*. Different letters (a, b, c) represent significant (*p* < 0.05) differences among different groups; the same letters indicate no significant differences between the two groups.

**Figure 5 ijms-25-09655-f005:**
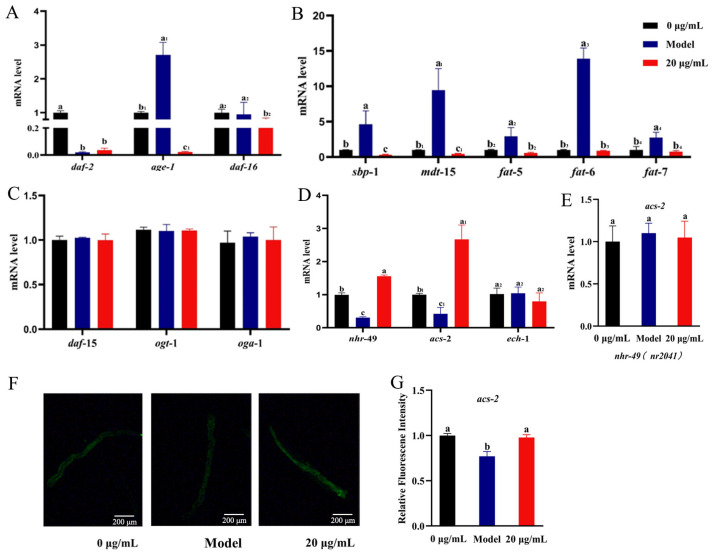
CSw reduce body fat content in *C. elegans* by NHR-49. (**A**) mRNA levels of genes related to insulin signalling (IIS) pathway; (**B**) mRNA levels of genes related to SBP-1/MDT-15 signalling pathway; (**C**) mRNA levels of genes related to TOR and hexosamine signalling pathway; (**D**) mRNA levels of genes related to NHR-49 signalling pathway; (**E**) expression of *acs-2* in *nhr-49* mutants after CSw treatment; (**F**,**G**) transgenic representative pictures and quantitative fluorescence intensity of *acs-2::GFP* in *C. elegans*. Different letters (a, b, c) represent significant (*p* < 0.05) differences among different groups; the same letters indicate no significant differences between the two groups. The same subscript number (a_1_, b_1_, c_1_, a_2_, b_2_, a_3_, b_3_, a_4_, b_4_) indicates that the significant differences analysis is performed within the same group.

**Figure 6 ijms-25-09655-f006:**
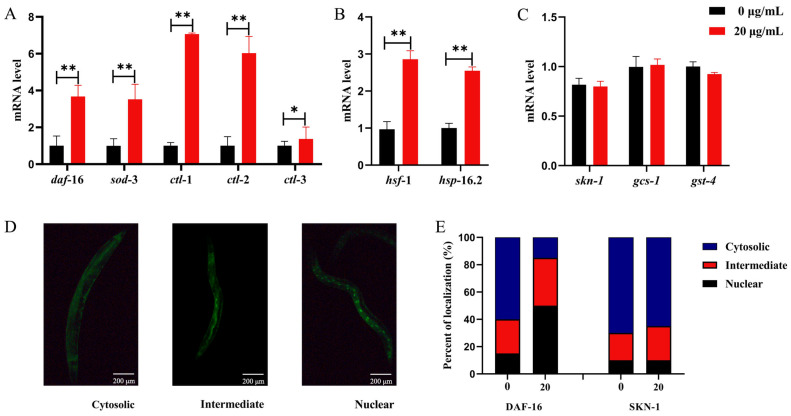
Life-prolonging effect of CSw depends on DAF-16 and HSF-1. (**A**–**C**) mRNA levels of *daf-16*, *hsf-1*, and *skn-1* and their target genes; (**D**,**E**) representative pictures and distribution of *daf-16* and *skn-1*. * *p* < 0.05, ** *p* < 0.01.

**Figure 7 ijms-25-09655-f007:**
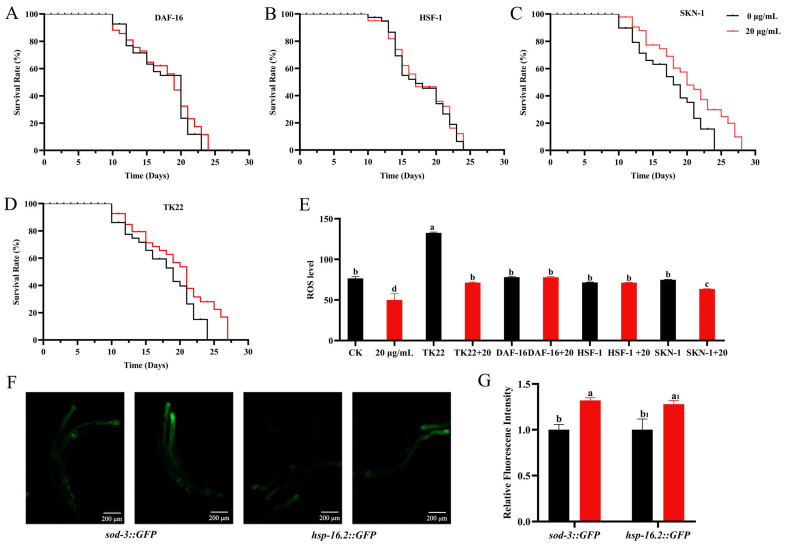
Life-prolonging effect of CSw depends on DAF-16 and HSF-1. (**A**–**D**) Survival curves of mutants *daf-16*, *hsf-1*, *skn-1*, and *mev-1*; (**E**) ROS levels of mutants treated with different concentrations of CSw; (**F**,**G**) transgenic representative pictures and quantitative fluorescence intensities of *sod-3::GFP* and *hsp-16.2::GFP* in *C. elegans*. Different letters (a, b, c, d) represent significant (*p* < 0.05) differences among different groups; the same letters indicate no significant differences between the two groups. The same subscript number (a_1_, b_1_) indicates that the significant differences analysis is performed within the same group.

**Table 1 ijms-25-09655-t001:** Preliminary identification of derivatives extracted from CSw.

No.	ID	Formula	Identification
1	M197T422	C_9_H_10_O_5_	Syringic acid
2	M359T569	C_18_H_16_O_8_	Rosmarinic acid
3	M118T126	C_15_H_24_O	Caryophyllene alpha-oxide
4	M118T126	C_5_H_11_NO_2_	Betaine
5	M439T818	C_30_H_48_O_3_	Oleanolic acid

## Data Availability

The data supporting the findings of this study have been previously published in the master’s thesis of the first author.

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
