# Peer review of "Clerodendranthus spicatus (Thunb.) Water Extracts Reduce Lipid Accumulation and Oxidative Stress in the Caenorhabditis elegans"

_ijms, 2024, doi:10.3390/ijms25179655_

Round 1
Reviewer 1 Report
Comments and Suggestions for Authors
The manuscript titled “Clerodendranthus spicatus (Thunb.) water extracts reduce lipid accumulation and oxidative stress in the Caenorhabditis elegans” by Xiao, X.; et al. is a scientific work where the authors studied the positive effects of Clerodendranthus spicatus water extract as antioxidative and antiageing compound. For it, Caenorhabditis elegans was used as model to visualize the regulation signal pathways to regulate the life spans and organism health. Many complementary techniques were devoted in this reasearch. This is an interesting topic and the manuscript is generally well-written. However, it exists some points that need to be addressed (please, see them below detailed point-by-point) to improve the scientific quality of the submitted manuscript paper before this article will be consider for its publication in the International Journal of Molecular Sciences.
1) The authors should consider to add the term “kidney tea” in the keyword list.
2) “Aging is an inevitable neurodegenerative process (…) hypertension, cancer and metabolic syndrome” (lines 29-31). Could the authors provide quantitative insights about the worldwide global burdens of those diseases associated to neurodegenerative processes? This could significantly aid the potential readers to better understand the research devoted in this work.
3) “For research on aging, C. elegans (…) IGF-1 (IIS) is the first definity longevity pathway identified (…) lifespan-related factors, stress-related factors, metabolism-related factors (…) antioxidant damage and prolonging lifespan in C. elegans is not known. (lines 63-75). Here, even if I agree with the information provided in these statements it should be also discussed how the nanoscale mechanical properties [1] can be decreased with the ageing of C. elegans nematodes according to the reduction of IGF-1 signaling pathway [2]. Thus, the cuticle stiffness could be used as a biomarker for healthspan.
[1] Magazzù, A.; et al. Investigation of Soft Matter Nanomechanics by Atomic Force Microscopy and Optical Tweezers: A Comprehensive Review. Nanomaterials 2023, 13, 963. https://doi.org/10.3390/nano13060963.
[2] Rahimi, M.; et al. Novel elasticity measurements reveal C. elegans cuticle stiffens with age and in a long-lived mutant. Biophys. J. 2022, 121, 515-524. https://doi.org/10.1016/j.bpj.2022.01.013.
4) Figure 1, panel C (line 96). The lateral scale bar should be added to the respective fluorescence microscopy images. Same comment for the Fig. 3, panel D (line 140), Fig. 4, panel A (line 156), Fig. 5, panel F (line 188), Fig. 6, panel D (line 218) and Fig. 7, panel F (line 242).
5) “An excellent anti-aging product should not only extend the lifespan but also keep the body healthy (…) health parameters of C. elegans, (…) detection index of C.elegans motor ability” (lines 108-111). Here, the authors assessed the reproduction rate, the degree of mobility, metabolism homeostasis and the heat shock resistance as key nematode health parameters. Did the authors carry out hypoxia resistance test to observe the ability of this organism to survive in low oxygen environments? Some information should be furnished in this regard.
6) Figure 3, panel A (line 140). The standard deviation bars should be added for each examined condition (the experiments according to measure the survival rates of C.elegans under the exposure of different Clerodendranthus spicatus water extract concentrations were conducted in triplicate as mentioned in the respective M&M section).
7) Discussion (lines 261-358). This section perfectly remarks the most relevant outcomes found by the authors in this work. No actions are requested from the authors.
8) “4.10 Nuclear translocation assay” (lines 481-489). Were the nuclear translocation assay measurements carried out with fixed worms or live organisms? This comment is based because fixation procedures could lead to the alteration of protein localization and eventually create artifacts in the study of real-time dynamic processes. Some information should be provided in this regard.
9) Conclusions (lines 533-541). It may be desirable to add a brief statement to discuss about the potential future action lines to pursue the topic covered in this research.
Comments on the Quality of English LanguageThe manuscript is generally well-written albeit it may be desirable if the authors could recheck it in order to polish some final details susceptible to be improved.
Author Response
Dear reviewer:
We feel great thanks for your professional review work on our manuscript “Clerodendranthus spicatus (Thunb.) water extracts reduce lipid accumulation and oxidative stress in the Caenorhabditis elegans”. These comments are very valuable and helpful for the revision and improvement of our paper. We have carefully studied the opinions and made corrections. The reviewer comments are laid out below and specific concerns have been numbered. Our modifications in the paper have been marked in red, and we hope to get the approval. The following is the reply to the comments of the three reviewers:
Comments and Suggestions for Authors
The manuscript titled “Clerodendranthus spicatus (Thunb.) water extracts reduce lipid accumulation and oxidative stress in the Caenorhabditis elegans” by Xiao, X.; et al. is a scientific work where the authors studied the positive effects of Clerodendranthus spicatus water extract as antioxidative and antiageing compound. For it, Caenorhabditis elegans was used as model to visualize the regulation signal pathways to regulate the life spans and organism health. Many complementary techniques were devoted in this reasearch. This is an interesting topic and the manuscript is generally well-written. However, it exists some points that need to be addressed (please, see them below detailed point-by-point) to improve the scientific quality of the submitted manuscript paper before this article will be consider for its publication in the International Journal of Molecular Sciences.
1) The authors should consider to add the term “kidney tea” in the keyword list.
Response: Thank you for your professional guidance. We have added the term “kidney tea” in the keyword list. Line 25 in the revised version.
2) “Aging is an inevitable neurodegenerative process (…) hypertension, cancer and metabolic syndrome” (lines 29-31). Could the authors provide quantitative insights about the worldwide global burdens of those diseases associated to neurodegenerative processes? This could significantly aid the potential readers to better understand the research devoted in this work.
Response: Thank you for your valuable comments. We have added more references on quantitative insights about the worldwide global burdens of those diseases associated to neurodegenerative processes into the INTRODUCTION part in the revised manuscript. Lines 31–34 in the revised version.
In a previous investigation, an analysis of data from multiple databases showed that metabolic syndrome was associated with a 2-fold increased risk of developing cardio-vascular disease and stroke, and a 1.5-fold increased risk of all-cause mortality [2, 3].
[2] Bozkurt, B., D. Aguilar, A. Deswal, S.B. Dunbar, G.S. Francis, T. Horwich, M. Jessup, M. Kosiborod, A.M. Pritchett, K. Ramasubbu, C. Rosendorff, and C. Yancy, Contributory Risk and Management of Comorbidities of Hypertension, Obesity, Diabetes Mellitus, Hyperlipidemia, and Metabolic Syndrome in Chronic Heart Failure: A Scientific Statement From the American Heart Association. Circulation, 2016. 134(23): e535-e578. http://dx.doi.org/10.1161/cir.0000000000000450
[3] Mottillo, S., K.B. Filion, J. Genest, L. Joseph, L. Pilote, P. Poirier, S. Rinfret, E.L. Schiffrin, and M.J. Eisenberg, The metabolic syndrome and cardiovascular risk a systematic review and meta-analysis. J Am Coll Cardiol, 2010. 56(14): 1113-32. http://dx.doi.org/10.1016/j.jacc.2010.05.034
3) “For research on aging, C. elegans (…) IGF-1 (IIS) is the first definity longevity pathway identified (…) lifespan-related factors, stress-related factors, metabolism-related factors (…) antioxidant damage and prolonging lifespan in C. elegans is not known. (lines 63-75). Here, even if I agree with the information provided in these statements it should be also discussed how the nanoscale mechanical properties [1] can be decreased with the ageing of C. elegans nematodes according to the reduction of IGF-1 signaling pathway [2]. Thus, the cuticle stiffness could be used as a biomarker for healthspan.
[1] Magazzù, A.; et al. Investigation of Soft Matter Nanomechanics by Atomic Force Microscopy and Optical Tweezers: A Comprehensive Review. Nanomaterials 2023, 13, 963. https://doi.org/10.3390/nano13060963.
[2] Rahimi, M.; et al. Novel elasticity measurements reveal C. elegans cuticle stiffens with age and in a long-lived mutant. Biophys. J. 2022, 121, 515-524. https://doi.org/10.1016/j.bpj.2022.01.013.
Response: Thank you for your introduction to these wonderful research work. According to your suggestion, we properly cite these article as:
A recent study showed that collagens are required for IIS signalling mutants' long lifespan and that overexpression of specific collagens extends wild-type lifespan [30, 31].
[30] Magazzù, A. and C. Marcuello, Investigation of Soft Matter Nanomechanics by Atomic Force Microscopy and Optical Tweezers: A Comprehensive Review. Nanomaterials, 2023. 13(6): 963. http://dx.doi.org//10.3390/nano13060963.
[31] Rahimi, M., S. Sohrabi, and C.T. Murphy, Novel elasticity measurements reveal C. elegans cuticle stiffens with age and in a long-lived mutant. Biophysical Journal, 2022. 121(4): 515-524. http://dx.doi.org/https://doi.org/10.1016/j.bpj.2022.01.013.
Lines 317-319 in the revised version.
4) Figure 1, panel C (line 96). The lateral scale bar should be added to the respective fluorescence microscopy images. Same comment for the Fig. 3, panel D (line 140), Fig. 4, panel A (line 156), Fig. 5, panel F (line 188), Fig. 6, panel D (line 218) and Fig. 7, panel F (line 242).
Response: Thank you for your careful checks. We have added the lateral scale bar to the images of Figure 1 (line 106), Figure 3 (line 149), Figure 4 (line 179), Figure 5 (line 193), Figure 6 (line 230) and Figure 7 (line 253) in the revised version.
5) “An excellent anti-aging product should not only extend the lifespan but also keep the body healthy (…) health parameters of C. elegans, (…) detection index of C.elegans motor ability” (lines 108-111). Here, the authors assessed the reproduction rate, the degree of mobility, metabolism homeostasis and the heat shock resistance as key nematode health parameters. Did the authors carry out hypoxia resistance test to observe the ability of this organism to survive in low oxygen environments? Some information should be furnished in this regard.
Response: Thank you for your professional guidance. We agree that hypoxia resistance test to observe the ability of C. elegans to survive in low oxygen environments would be useful to understand details of health enhancement of nematodes. At this point, we do not have the necessary tool-set to study the proposed hypoxic tolerance test. We hope that in future studies we can use hypoxia tolerance tests to further evaluate the stress resistance of C. elegans.
6) Figure 3, panel A (line 140). The standard deviation bars should be added for each examined condition (the experiments according to measure the survival rates of C.elegans under the exposure of different Clerodendranthus spicatus water extract concentrations were conducted in triplicate as mentioned in the respective M&M section).
Response: Thank you for your valuable comments. For Stress resistance assays, according to your suggestion, we have re-organized and analyzed the data. The survival curves were analyzed using Kaplan-Meier and p-values were calculated using the log-rank test. Figure 3 (Line149) in the revised version.
7) Discussion (lines 261-358). This section perfectly remarks the most relevant outcomes found by the authors in this work. No actions are requested from the authors.
Response: Thank you for your careful reading.
8) “4.10 Nuclear translocation assay” (lines 481-489). Were the nuclear translocation assay measurements carried out with fixed worms or live organisms? This comment is based because fixation procedures could lead to the alteration of protein localization and eventually create artifacts in the study of real-time dynamic processes. Some information should be provided in this regard.
Response: Thank you for your question, according to your comments. We have described this part in more detail. Lines 506-513 in the revised version.
Nuclear translocation assay was performed as described [23, 46]. At the L4 stage, the worms TJ356 (daf-16::GFP) and LD1 (skn-1::GFP) were transferred to NGM culture plates which included 20 μg/mL CSw and standard NGM as a control. After 7 days of incubation at 20 â—¦C, the nematodes were picked onto agar pads and anaesthetised with 10 µL of levamisole hydrochloride solution at a concentration of 10 mmol/L. With a microscope, the nematodes skn-1::GFP and daf-16::GFP showed changes in their fluo-rescence. GFP expression patterns and defined as three types: “cytosolic”, “intermedi-ate” and “nuclear”. The distribution rate was calculated in three separate experiments with at least 30 individuals each time.
[23] Liu, M., N. Li, X. Lu, S. Shan, X. Gao, Y. Cao, and W. Lu, Sweet tea (Rubus Suavissmus S. Lee) polysaccharides promote the longevity of Caenorhabditis elegans through autophagy-dependent insulin and mitochondrial pathways. Int J Biol Macromol, 2022. 207: 883-892. http://dx.doi.org/10.1016/j.ijbiomac.2022.03.138.
[46] Li, S.-T., H.-Q. Zhao, P. Zhang, C.-Y. Liang, Y.-P. Zhang, A.-L. Hsu, and M.-Q. Dong, DAF-16 stabilizes the aging transcriptome and is activated in mid-aged Caenorhabditis elegans to cope with internal stress. Aging Cell, 2019. 18(3): e12896. http://dx.doi.org/https://doi.org/10.1111/acel.12896.
9) Conclusions (lines 533-541). It may be desirable to add a brief statement to discuss about the potential future action lines to pursue the topic covered in this research.
Response: Thank you for your suggestion. According to your suggestion, we have added to the conclusion. Because the purpose of our study was to know the safety and health promotion effects of Clerodendranthus spicatus as a dietary supplement, from the results obtained so far, it is safe. However, we know that foods with well-known antioxidant attractions can produce antioxidant effects, so, CSw as a daily drinking tea requires more safety evaluations should be carried out in other mammalian experiments, such as in mice, and standards should be set for the recommended doses. And when drinking Clerodendranthus spicatus daily, the dosage should be carefully controlled until a conclusion about their safety is reached. Lines 563-565 in the revised version.
Comments on the Quality of English Language
The manuscript is generally well-written albeit it may be desirable if the authors could recheck it in order to polish some final details susceptible to be improved.
Response: we tried our best to improve the manuscript and made some changes to the manuscript. These changes will not influence the content and framework of the paper. We appreciate for reviewer‘s warm work and hope that the correction will meet with approval.
Reviewer 2 Report
Comments and Suggestions for Authors
The objective of this study was to investigate the effects of Clerodendranthus spicatus (Kidney Tea) on the longevity and health of Caenorhabditis elegans (C. elegans), evaluating its antioxidant and hypolipidemic properties and its potential to extend lifespan, in order to understand its safety and potential as a healthy dietary supplement. The topic of the article is relevant and could contribute to research on the beneficial properties of Clerodendranthus spicatus. However, I believe the study lacks a clearly defined objective that aligns with the content of the article, and the proposed objective does not correspond to the title. Although the conclusion is supported by the results, I suggest considering the publication of this work only after the following major revisions have been made:
- Line 74: The phrase "antioxidant damage" seems to be incorrectly used. The authors likely intended to say "oxidative damage."
- Abbreviations: It is essential to define all abbreviations used, such as DAF-16, NRF2, and 6-OHDA, among many others, to avoid confusion.
- When assessing lipofuscin levels, the number of experiments performed is mentioned, but the number of nematodes used is not. This information should be specified for clarity.
- Line 441: The probe should be defined and written in uppercase as H2-DCF-DA.
- NGM Preparation: It is important to detail how NGM is prepared for maintaining the worms. Later, the addition of cholesterol is mentioned, but standard NGM recipes already include cholesterol. Is the same amount used? This should be clarified.
- Triglyceride, free fatty acid, and real-time quantitative PCR assays: In sections 8.2, 4.8.3, and 4.9, respectively, it is not mentioned whether nematode lysis was performed or the method used. Additionally, it is necessary to indicate that the gene sequences are provided in the supplementary materials.
- Lifespan graphs: Were Kaplan-Meier methods used to create the lifespan graphs? Only the log-rank test (Mantel-Cox) is mentioned, but this test is used to assess statistical differences. It is important to specify the graphing method used.
- Statistical analysis: It is mentioned that the results are presented as means ± standard deviation, but most data are reported with standard error. Additionally, it is not stated how the normality test was conducted or why an ANOVA was chosen.
- Figure 1a: In the lifespan analysis, it is mentioned that the 20 and 100 mg/mL concentrations have the most significant effect. However, the graph does not show that 100 mg/mL is better; it appears that 50 mg/mL is more effective. For better visualization, I suggest adding a table summarizing the lifespan results, including the mean, limits, and maximum lifespan. They could refer to the table in the article by Hernández-Cruz et al. (2024) in Antioxidants.
- Images: The magnification used to take the photographs and the scale at which C. elegans is shown are not mentioned. This is crucial for correctly interpreting the images.
- Figure 2: It is not indicated which group corresponds to each color.
- Graph quality: Most graphs have very low quality and appear blurry. It is recommended to improve their resolution for better interpretation of the results. The letters are not distinguishable.
- Figure 3 (a, b, and c): These should be made using the Kaplan-Meier method. Additionally, all abbreviations used need to be defined.
- Figure 4: It is placed in the wrong section and needs to be reorganized.
- Meaning of the bars: It is unclear what "ab" or "bc" on the significance bars represent.
- Contradictory results: An increase in body length and a decrease in body width in the nematodes are mentioned, but only the increase in length is clearly observed. Additionally, it is stated that these results led to evaluating lipid accumulation, but the protocol indicates it is a cholesterol-induced obesity model, suggesting that the goal is to evaluate the effect of the extract on obesity in C. elegans. This objective is not clearly stated from the beginning, so the authors are asked to be consistent and maintain the same narrative from the introduction. Moreover, in the obesity model, it is not mentioned why only the 20 mg/mL concentration was selected for further evaluations.
- Nuclear translocation images: The difference between cytosolic and intermediate classification is not clearly visible. Improving the quality of these images would be helpful for clearer interpretation.
- Discussion: The first paragraph of the discussion does not belong to this section; it seems to be part of the instructions for authors and should be removed.
· Line 36. Discuss the harm caused by consuming botanical supplements. You need to include some research as an example.
· Line 40 Using both common and scientific names can be confusing and does not add value to the text. It is necessary to use only the name used in this research and mention that it is known by other names.
· Line 46. Does kidney tea consist of other plants? Do they have the same properties?
· Line 52. The phrase “anti-oxidative stress” seems to be incorrectly used.
· Line 66. Even fluorescent markers need a citation.
· Line 72. What is the importance of the genes that are activated?
· The chromatograms provided are unclear and overly saturated with data. Perhaps presenting the compounds in a table would be more suitable
· It is necessary to include the abbreviated form of the plant's scientific name, which should be written in italics. The genus name is abbreviated to its initial letter in uppercase, followed by a period, and the species name in lowercase. Example: C. elegans and C. spicatus
Author Response
Dear reviewer:
We feel great thanks for your professional review work on our manuscript “Clerodendranthus spicatus (Thunb.) water extracts reduce lipid accumulation and oxidative stress in the Caenorhabditis elegans”. These comments are very valuable and helpful for the revision and improvement of our paper. We have carefully studied the opinions and made corrections. The reviewer comments are laid out below and specific concerns have been numbered. Our modifications in the paper have been marked in red, and we hope to get the approval. The following is the reply to the comments of the three reviewers:
Comments and Suggestions for Authors
The objective of this study was to investigate the effects of Clerodendranthus spicatus (Kidney Tea) on the longevity and health of Caenorhabditis elegans (C. elegans), evaluating its antioxidant and hypolipidemic properties and its potential to extend lifespan, in order to understand its safety and potential as a healthy dietary supplement. The topic of the article is relevant and could contribute to research on the beneficial properties of Clerodendranthus spicatus. However, I believe the study lacks a clearly defined objective that aligns with the content of the article, and the proposed objective does not correspond to the title. Although the conclusion is supported by the results, I suggest considering the publication of this work only after the following major revisions have been made:
1) Line 74: The phrase "antioxidant damage" seems to be incorrectly used. The authors likely intended to say "oxidative damage."
Response: Thank you for your careful reading. We have corrected the “antioxidant damage” into “oxidative damage”. Lines 807 – 808 in the revised version.
2) Abbreviations: It is essential to define all abbreviations used, such as DAF-16, NRF2, and 6-OHDA, among many others, to avoid confusion.
Response: Thank you for your careful checks. We re-checked the manuscript and all abbreviations were stated when they first appeared. Lines 20-21, 76, 188 in the revised version.
3) When assessing lipofuscin levels, the number of experiments performed is mentioned, but the number of nematodes used is not. This information should be specified for clarity.
Response: Thank you for your careful checks. According to your suggestion, we have made the following modifications.
For the lipofuscin assay, at least 30 nematodes were photographed in each group, we have added this information in the manuscript. Lines 409 - 410 in the revised version.
4) Line 441: The probe should be defined and written in uppercase as H2-DCF-DA.
Response: Thank you for your professional guidance. We have made corresponding changes in the manuscript.
Using a previously reported method, reactive oxygen species (ROS) levels in C. elegans were measured using 1 μM H2-DCF-DA (DCFH-DA). Line 456 in the revised version.
5) NGM Preparation: It is important to detail how NGM is prepared for maintaining the worms. Later, the addition of cholesterol is mentioned, but standard NGM recipes already include cholesterol. Is the same amount used? This should be clarified.
Response: Thank you for your careful reading. According to your suggestion, we have made the following modifications.
Preparation of the CSw medium: Combined with the NGM preparation method, 5 mg/ mL CSw reserve solution was prepared, dissolved in sterile water, refrigerated for use, diluted into the required concentration solution when needed, and the amount of other substances added remained unchanged.
Preparation of high cholesterol medium: Combined with the NGM preparation method, the amount of cholesterol was changed to 1.25 mL (the amount of cholesterol in 250 mL ordinary NGM was 0.25 mL), and the amount of other substances added remained unchanged. Lines 387-394 in the revised version.
6) Triglyceride, free fatty acid, and real-time quantitative PCR assays: In sections 4.8.2, 4.8.3, and 4.9, respectively, it is not mentioned whether nematode lysis was performed or the method used. Additionally, it is necessary to indicate that the gene sequences are provided in the supplementary materials.
Response: Thank you for your careful checks. we have made corresponding changes in the manuscript.
- elegans(0.1 g) to be tested were washed into EP tube with M9 buffer on the 8th day. The triglyceride quantification kit (Solarbio, Beijing, China) was used to determine the triglyceride content by adding 1 mL Reagent 1 into an EP tube, ice bath homogeniza-tion, 8000 g, centrifugation at 4℃ for 10 min, and taking the supernatant solution for determination. Lines 478-482, 486-490 in the revised version.
7) Lifespan graphs: Were Kaplan-Meier methods used to create the lifespan graphs? Only the log-rank test (Mantel-Cox) is mentioned, but this test is used to assess statistical differences. It is important to specify the graphing method used.
Response: Thank you for your professional guidance. Kaplan-Meier methods used to create the lifespan graphs. we have made corresponding changes in the manuscript.
The log rank (Mantel-Cox) test was used to determine statistical significance for the life span analysis. The Kaplan-Meier method was used to create the lifespan graphs. Lines 549-550 in the revised version.
8) Statistical analysis: It is mentioned that the results are presented as means ± standard deviation, but most data are reported with standard error. Additionally, it is not stated how the normality test was conducted or why an ANOVA was chosen.
Response: Thank you for your professional guidance. According to your suggestion, we have made the following modifications.
The statistical significance was evaluated using one-way analysis of variance (ANOVA) followed by Duncan's method for multiple comparisons with SPSS22.0. Different let-ters (a, b, c, d) represent significant (p < 0.05) differences among different groups. Lines 550-553 in the revised version.
9) Figure 1a: In the lifespan analysis, it is mentioned that the 20 and 100 mg/mL concentrations have the most significant effect. However, the graph does not show that 100 mg/mL is better; it appears that 50 mg/mL is more effective. For better visualization, I suggest adding a table summarizing the lifespan results, including the mean, limits, and maximum lifespan. They could refer to the table in the article by Hernández-Cruz et al. (2024) in Antioxidants.
Response: Thank you for your careful reading. Due to our carelessness, there is a mistake in the expression, we have made corresponding changes in the manuscript, and the form is added to the supplementary.
Table S1 Effects of CSw on the lifespan of C. elegans.
|
Strain(solvent) |
Maximum lifespan(d) |
Mean lifespan (d)(±SEM) |
p value versus control |
|
N2 (Control) |
23 |
17.00±0.89 |
|
|
N2 (20 μg/mL) |
29 |
21.46±0.71 |
<0.01 |
|
N2 (50 μg /mL ) |
28 |
20.63±0.77 |
<0.05 |
|
N2 (100 μg / mL ) |
26 |
19.40±0.73 |
|
|
N2 (500 μg / mL ) |
24 |
18.46±0.62 |
|
|
N2 (1000 μg / mL ) |
25 |
18.33±0.88 |
|
10) Images: The magnification used to take the photographs and the scale at which C. elegans is shown are not mentioned. This is crucial for correctly interpreting the images.
Response: Thank you for your careful checks. We have added the lateral scale bar to the images of Figure 1 (line 105), Figure 3 (line 149), Figure 4 (line 179), Figure 5 (line 192), Figure 6 (line 229) and Figure 7 (line 252) in the revised version.
11) Figure 2: It is not indicated which group corresponds to each color.
Graph quality: Most graphs have very low quality and appear blurry. It is recommended to improve their resolution for better interpretation of the results. The letters are not distinguishable.
Response: Thank you for your careful reading. According to your suggestion, we have reworked the image and uploaded a higher resolution image. Figure 2 (line 128) in the revised version.
12) Figure 3 (a, b, and c): These should be made using the Kaplan-Meier method. Additionally, all abbreviations used need to be defined.
Response: Thank you for your careful checks. According to your suggestion, we have reworked the image, and the Kaplan-Meier method was used to create the lifespan graphs. Additionally, all abbreviations used are defined. Figure 3 (line 149) in the revised version.
13) Figure 4: It is placed in the wrong section and needs to be reorganized.
Response: Thank you for your careful reading. Due to our carelessness, Figure 4 was placed in the wrong section, we have made corresponding changes in the manuscript. Figure 4 (line 179) in the revised version.
14) Meaning of the bars: It is unclear what "ab" or "bc" on the significance bars represent.
Contradictory results: An increase in body length and a decrease in body width in the nematodes are mentioned, but only the increase in length is clearly observed. Additionally, it is stated that these results led to evaluating lipid accumulation, but the protocol indicates it is a cholesterol-induced obesity model, suggesting that the goal is to evaluate the effect of the extract on obesity in C. elegans. This objective is not clearly stated from the beginning, so the authors are asked to be consistent and maintain the same narrative from the introduction. Moreover, in the obesity model, it is not mentioned why only the 20 mg/mL concentration was selected for further evaluations.
Nuclear translocation images: The difference between cytosolic and intermediate classification is not clearly visible. Improving the quality of these images would be helpful for clearer interpretation.
Discussion: The first paragraph of the discussion does not belong to this section; it seems to be part of the instructions for authors and should be removed.
Response: Thank you for your valuable comments. According to your suggestion, we have reworked the image and uploaded a higher resolution image. The results are described in more detail. The specific changes are as follows:
Different letters (a, b, c, d) represent significant (p < 0.05) differences among different groups. ab means no significant difference from a, b. In order to better understand the significant differences, we have added more details in the RESULTS.
As illustrated in Figures 2 C-D, 100 μg/mL CSw significantly increased the body length and reduced the body width of C. elegans, and 100 μg/mL CSw significantly enhanced the movement of both young and old nematodes (Figures 2 E-F). As shown in the re-sults above, CSw enhanced nematode health indicators. Lines 124-126 in the revised version.
In a study of C. spicatus reducing appetite and reducing fat accumulation, C. spicatus was found to reduce food intake and visceral fat mass in rats [14].
[14] Son, J.-Y., S.-Y. Park, J.-Y. Kim, K.-C. Won, Y.-D. Kim, Y.-J. Choi, M.S. Zheng, J.-K. Son, and Y.-W. Kim, Orthosiphon stamineus reduces appetite and visceral fat in rats. J Korean Soc Appl Bi, 2011. 54(2): 200-205. http://dx.doi.org/10.3839/jksabc.2011.033
Research on the lipid-lowering effect of C. spicatus was added in the INTRODUCTION. Because the lipid-lowering effect is considered to be included in the health-promoting effect, there is no more expression on lipid-lowering in the introduction. The signaling pathway of lipid-lowering is expressed in more detail in the DISCUSSION. We hope that the correction will meet with approval. Lines 57-59 in the revised version.
Considering that 20 μg/mL CSw-treat group had the best life prolongation effect on C. elegans and showed certain lipid lowering effect in the previous fat deposition experiment, this concentration was selected to continue the subsequent experiment. Lines 183-185 in the revised version.
15) Line 36. Discuss the harm caused by consuming botanical supplements. You need to include some research as an example.
Response: Thank you for your suggestion. we have made corresponding changes in the manuscript.
For example, the improper use of He-Shou-Wu [Reynoutria multiflora (Thunb.) Mold-enke] and Huang-Yao-Zi (Dioscorea bulbifera L.) have caused liver damage [6].
[6] Zhang, P., Y. Ye, X. Yang, and Y. Jiao, Systematic Review on Chinese Herbal Medicine Induced Liver Injury. Evidence-Based Complementary and Alternative Medicine, 2016. 2016: 3560812. http://dx.doi.org/10.1155/2016/3560812
Lines 40-42 in the revised version.
16) Line 40 Using both common and scientific names can be confusing and does not add value to the text. It is necessary to use only the name used in this research and mention that it is known by other names.
Response: thank you for your professional guidance. According to your suggestion, we have made corresponding changes in the manuscript.
C. spicatusis also known as Orthosiphon aristatus andOrthosiphon stamineuso and refers to "kidney tea" in literature and pharmacopoeia. Lines 50-51 in the revised version.
17) Line 46. Does kidney tea consist of other plants? Do they have the same properties?
Response: Thank you for the question. Clerodendranthus spicatus, Orthosiphon aristatus and Orthosiphon stamineuso all refer to the same tea, although they have different Latin names, and are generally translated into English from kidney tea.
18) Line 52. The phrase “anti-oxidative stress” seems to be incorrectly used.
Response: Thank you for your careful reading. We have corrected the “anti-oxidative stress” into “oxidative stress”. Lines 54-55 in the revised version.
19) Line 66. Even fluorescent markers need a citation.
Response: Thank you for your suggestion. we have made corresponding changes in the manuscript.
For example, the eggs, pharynx, gut, muscle cells and embryonic cells of nematode worms are easily observed. And it also can be used to observe fluorescent markers [18].
[18] Johnson, T.E., Advantages and disadvantages of Caenorhabditis elegans for aging research. Exp Gerontol, 2003. 38(11): 1329-1332. http://dx.doi.org/https://doi.org/10.1016/j.exger.2003.10.020
Lines 72-74 in the revised version.
20) Line 72. What is the importance of the genes that are activated?
Response:·Thank you for the question. we have added more details about this part in the manuscript.
When IIS signalling is inhibited, DAF-16 and SKN-1 can generate nuclear translocation and activate the expression of downstream target genes, and affect the life span and response to various kinds of stress in C. elegans.
Lines 79-81 in the revised version.
21) The chromatograms provided are unclear and overly saturated with data. Perhaps presenting the compounds in a table would be more suitable
Response:·Thank you for your valuable comments. we have added the form to the supplementary.
For the components in CSw, positive and negative chromatograms were obtained, where a total of 43 molecules were identified (supplementary Table S2, Figures. 8 A-B).
22) It is necessary to include the abbreviated form of the plant's scientific name, which should be written in italics. The genus name is abbreviated to its initial letter in uppercase, followed by a period, and the species name in lowercase. Example: C. elegans and C. spicatus.
Response: Thank you for your suggestion. We went through the entire manuscript and revised the “Caenorhabditis elegans” and “Clerodendranthus spicatus” into “C. elegans” and “C. spicatus”.
Reviewer 3 Report
Comments and Suggestions for Authors
Review of the paper entitled “Clerodendranthus spicatus (Thunb.) water extracts reduce lipid accumulation and oxidative stress in the Caenorhabditis elegans” by Xian Xiao, Fan-hua Wu, Bing Wang, Ze-ping Cai, Lan-ying Wang, Yun-fei Zhang, Xu-dong Yu and Yan-ping Luo
The results obtained by the Authors are interesting and quite promising… . Nevertheless, I have a few comments.
Since not all who will read the authors’ paper use the model organism Caenorhabditis elegans (C. elegans) in their research, sentences like “from the beginning of the L4 stage until day 8. “C. elegans were cultured using the Brenner method at 20 °C. Wild-type N2 and uracil-deficient E. coli OP50” will not be fully clear to these readers.
Thus, I suggest the authors add a few, a dozen or so sentences describing the anatomy, food, life cycle, sexual forms etc. of C. elegans. The authors could consider presenting the life cycle using a scheme, figure. This will help readers better understand the authors' paper. After all, not everyone knows that the life cycle of this nematode from egg to egg-laying adult grass is only 3 days at a temperature of 25°C.
The authors used the wild-type (WT) Bristol N2 and several mutant strains of C. elegans in their experiments. If there are phenotypic differences between C. elegans mutants and WT strain and between individual mutants, it might be worth showing photos. It would be good if the authors briefly characterized these genes, i.e. daf-16, skn-1, mev-1, etc. What they are responsible for, what does the mutation in daf-16 (skn-1, mev-1, etc.) cause, whether it increases or reduces lifespan compared to that of the wild type, etc. This could be shown in a Table with appropriate literature references.
Results. Does the abbreviation CK stand for control? Which animals did the authors treat as control: WT Bristol N2 treated Clerodendranthus spicatus water extracts (CSw), WT Bristol N2 untreated CSw, the corresponding mutant treated CSw, the corresponding mutant untreated CSw…. ? This is not clearly shown. In general, the results obtained by the authors should be better described. The authors wrote for example: “The mean life span of C. elegans increased to 21.46 ± 0.71 compared with the control group”. But here again I repeat my question: which animals were the control group? Does this increase in life span concern the WT strain or the mutant strains. What do the letters a, b, c, ab, bc on the bars mean? Although the authors wrote "Bars with no letters in common are significantly different", it is not clear to me. For example Figure 1B. . The letter “a” is on both black bars (4d, 8d), but the values are different. Similarly with the letters “b” and “c”. The reader can indeed guess the authors' intentions, but it would be better if he did not have to guess but just knew. Similarly Figure 2D. The blue, red and orange bars have the letters "ab". Does this mean that the result is statistically significant only for the group receiving CSw at a concentration of 100µg/ml?
When presenting the results for the mutant C. elegans strains, the authors use two descriptors: Control and Model. Was there appropriate control for every mutant?
It is obvious that the results obtained in the control group are used for comparison with the results from the experimental group (groups). And only then can we obtain statistical certainty that the tested drugs, herbs, food etc. administered to the experimental group (groups) actually brought the expected effect. I repeat this question about the control group, because it is the most important in this type of experiment. The value of the obtained experimental results depends on the appropriate design of the control group (groups). Therefore, I kindly ask the authors to describe the control group (groups) in each experiment in a very detailed and precise manner.
What does „Model” mean?
The abbreviation CSw should be entered by the authors also in the main text, not only in the abstract. The authors should expand the abbreviation 6-OHDA.

Author Response
Dear reviewer:
We feel great thanks for your professional review work on our manuscript “Clerodendranthus spicatus (Thunb.) water extracts reduce lipid accumulation and oxidative stress in the Caenorhabditis elegans”. These comments are very valuable and helpful for the revision and improvement of our paper. We have carefully studied the opinions and made corrections. The reviewer comments are laid out below and specific concerns have been numbered. Our modifications in the paper have been marked in red, and we hope to get the approval. The following is the reply to the comments of the three reviewers:
Comments and Suggestions for Authors
Review of the paper entitled “Clerodendranthus spicatus (Thunb.) water extracts reduce lipid accumulation and oxidative stress in the Caenorhabditis elegans” by Xian Xiao, Fan-hua Wu, Bing Wang, Ze-ping Cai, Lan-ying Wang, Yun-fei Zhang, Xu-dong Yu and Yan-ping Luo
The results obtained by the Authors are interesting and quite promising… . Nevertheless, I have a few comments.
1) Since not all who will read the authors’ paper use the model organism Caenorhabditis elegans (C. elegans) in their research, sentences like “from the beginning of the L4 stage until day 8. “C. elegans were cultured using the Brenner method at 20 °C. Wild-type N2 and uracil-deficient E. coli OP50” will not be fully clear to these readers. Thus, I suggest the authors add a few, a dozen or so sentences describing the anatomy, food, life cycle, sexual forms etc. of C. elegans. The authors could consider presenting the life cycle using a scheme, figure. This will help readers better understand the authors' paper. After all, not everyone knows that the life cycle of this nematode from egg to egg-laying adult grass is only 3 days at a temperature of 25°C.
Response: Thank you for your valuable comments. According to your suggestion, we have made corresponding changes in the manuscript.
For research on aging, C. elegans is the preferred model. It is very easy to raise and the food is mainly E. coli OP50. Its life cycle is very fast, taking 12 hours to form an embryo, 60 hours to develop into an adult, and its lifespan is about 21 days. Moreover, it is possible to obtain plenty of nematodes that are all the same age, which enhances the reproducibility of studies involving vital analysis. The body is translucent, facili-tating the observation of the various tissues of the body. For example, the eggs, phar-ynx, gut, muscle cells and embryonic cells of nematode worms are easily observed. And it also can be used to observe fluorescent markers. Lines 67-74 in the revised version.
2) The authors used the wild-type (WT) Bristol N2 and several mutant strains of C. elegans in their experiments. If there are phenotypic differences between C. elegans mutants and WT strain and between individual mutants, it might be worth showing photos. It would be good if the authors briefly characterized these genes, i.e. daf-16, skn-1, mev-1, etc. What they are responsible for, what does the mutation in daf-16 (skn-1, mev-1, etc.) cause, whether it increases or reduces lifespan compared to that of the wild type, etc. This could be shown in a Table with appropriate literature references.
Response: Thank you for your suggestion. We agree that this will give a better understanding of the differences between wild type C. elegans and mutant strains, but we encourage to use the CGC website for more information, so we have added a note to the manuscript. All information about the mutant strains can be found on the GGC website. We hope that the note will meet with approval.
3) Results. Does the abbreviation CK stand for control? Which animals did the authors treat as control: WT Bristol N2 treated Clerodendranthus spicatus water extracts (CSw), WT Bristol N2 untreated CSw, the corresponding mutant treated CSw, the corresponding mutant untreated CSw…. ? This is not clearly shown. In general, the results obtained by the authors should be better described. The authors wrote for example: “The mean life span of C. elegans increased to 21.46 ± 0.71 compared with the control group”. But here again I repeat my question: which animals were the control group? Does this increase in life span concern the WT strain or the mutant strains. What do the letters a, b, c, ab, bc on the bars mean? Although the authors wrote "Bars with no letters in common are significantly different", it is not clear to me. For example Figure 1 B. The letter “a” is on both black bars (4d, 8d), but the values are different. Similarly with the letters “b” and “c”. The reader can indeed guess the authors' intentions, but it would be better if he did not have to guess but just knew. Similarly Figure 2 D. The blue, red and orange bars have the letters "ab". Does this mean that the result is statistically significant only for the group receiving CSw at a concentration of 100 µg/ml?
Response: Thank you for your valuable comments. According to your suggestion, we describe the results in more detail. All the figures were redrawn, with 0 μg/mL instead of control.
Compared with the 0 μg/mL CSw-treated group, other concentrations of CSw-treated did not inhibit the egg production of C. elegans and showed a facilitative effect. Lines 113-115 in the revised version.
As illustrated in Figures 2 C-D, 100 μg/mL CSw significantly increased the body length and reduced the body width of C. elegans, and 100 μg/mL CSw significantly en-hanced the movement of both young and old nematodes (Figures 2 E-F). Lines 124-126 in the revised version.
4) When presenting the results for the mutant C. elegans strains, the authors use two descriptors: Control and Model. Was there appropriate control for every mutant?
Response: Thank you for your question. According to your suggestion, we rework all the figures, with 0 μg/mL instead of control. And made corresponding changes in the manuscript.
Nematodes were cultured as described in Subsection 4.8.1, and high cholesterol medium was prepared as described in Subsection 4.1. Lines 166-168 in the revised version.
4.8.1 For obesity modeling, synchronous L4-stage worms were transferred to an NGM plate containing high cholesterol (model: +cholesterol). Some worms were grown on a standard NGM plate (with CSw 0 μg/mL). Other worms were moved to an NGM plate (+cholesterol with CSw 20, 50, 100 and 500 μg/mL).
The mRNA expression of nhr-49 was significantly up-regulated 1.55-fold in the 20 μg/mL CSw group compared with the model (+cholesterol) group. Lines 189-190 in the revised version.
5) It is obvious that the results obtained in the control group are used for comparison with the results from the experimental group (groups). And only then can we obtain statistical certainty that the tested drugs, herbs, food etc. administered to the experimental group (groups) actually brought the expected effect. I repeat this question about the control group, because it is the most important in this type of experiment. The value of the obtained experimental results depends on the appropriate design of the control group (groups). Therefore, I kindly ask the authors to describe the control group (groups) in each experiment in a very detailed and precise manner.
What does “Model” mean?
Response: Thank you for your valuable comments. we tried our best to improve the manuscript and as much as possible to avoid confusion between the treatment group and the control group. And all the figures were redrawn, with 0 μg/mL instead of control.
For obesity modeling, synchronous L4-stage worms were transferred to an NGM plate containing high cholesterol (model: +cholesterol). Some worms were grown on a standard NGM plate (with CSw 0 μg/mL). Other worms were moved to an NGM plate (+cholesterol with CSw 20, 50, 100 and 500 μg/mL). Lines 469-470 in the revised version.
6) The abbreviation CSw should be entered by the authors also in the main text, not only in the abstract. The authors should expand the abbreviation 6-OHDA.
Response: Thank you for your careful checks. We re-examined the entire manuscript and made corresponding changes in the manuscript. Lines 53 and 376 in the revised version.
Round 2
Reviewer 2 Report
Comments and Suggestions for Authors
After reviewing this work, I believe that most of the observations have been addressed. However, it is essential that the authors take the necessary time to correct the manuscript, as the responses appear to be hasty. For instance, in the first revision, the authors indicate that the adjustment was made in lines 807-808; however, this line does not exist in the text, and this error is repeated throughout the review. I suggest considering accept the publication after minor revisions:
• It is necessary to write the full name of the species or compound, followed by the abbreviation that will be used in the text. This abbreviation should be placed in parentheses.
• It is necessary to define the full name of the probe H2DCF-DA.
• In Figure 1, the scale bars are not distinguishable. This is crucial for correctly interpreting the images.
• If there are no statistically significant differences, it is not necessary to indicate them in the graphs. Labeling all the bars with letters complicates the analysis for the reader.
• The chromatogram is not a result as it represents unprocessed data. It is necessary to replace it with a properly analyzed composition table. The chromatogram can be included in the supplementary material.
• Nuclear translocation images: The difference between cytosolic and intermediate classification is not clearly visible. Improving the quality of these images would be helpful for clearer interpretation.
Author Response
Dear reviewer:
Thank you for your professional review of our article "Clerodendranthus spicatus (Thunb.) water extract reduce lipid accumulation and oxidative stress in the Caenorhabditis elegans ". We apologize for our carelessness, which has caused some problems in the modified version. our comments are very valuable for the revision and improvement of our paper. We carefully studied the opinions again and made amendments. The changes in the paper have been marked in red and we hope to get the approval.
Comments and Suggestions for Authors
After reviewing this work, I believe that most of the observations have been addressed. However, it is essential that the authors take the necessary time to correct the manuscript, as the responses appear to be hasty. For instance, in the first revision, the authors indicate that the adjustment was made in lines 807-808; however, this line does not exist in the text, and this error is repeated throughout the review. I suggest considering accept the publication after minor revisions:
1) It is necessary to write the full name of the species or compound, followed by the abbreviation that will be used in the text. This abbreviation should be placed in parentheses.
Response: Thank you for your valuable comments. According to your suggestion, we have reviewed the manuscript and put the abbreviation in parentheses. Lines 75, 149-151, 154, 157-158, 168, 172-173 and 185-186 in the revised version.
2) It is necessary to define the full name of the probe H2DCF-DA.
Response: Thank you for your careful reading. According to your suggestion, we have made corresponding changes in the manuscript.
H2-DCF-DA (dihydrodichlorofluorescein diacetate). Line 452 in the revised version.
3) In Figure 1, the scale bars are not distinguishable. This is crucial for correctly interpreting the images.
Response: Thank you for your careful checks. We re-adjusted the scale bars of Figure 1 and uploaded a clearer picture, line 104 in the revised version.
4) If there are no statistically significant differences, it is not necessary to indicate them in the graphs. Labeling all the bars with letters complicates the analysis for the reader.
Response: Thank you for your valuable comments. In order to give readers a clearer understanding of the significant differences in the data, we have added this sentence, " contains the same letters, indicating no significant differences between the two groups. " In the 4.13 Statistical analysis of the manuscript, we hope to get your approval.
Different letters (a, b, c, d) represent significant (p < 0.05) differences among different groups, and contains the same letters, indicating no significant differences between the two groups. Lines 547-549 in the revised version.
5) The chromatogram is not a result as it represents unprocessed data. It is necessary to replace it with a properly analyzed composition table. The chromatogram can be included in the supplementary material.
Response: Thank you for your professional guidance. According to your suggestion, we put the chromatogram into the supplementary materials and added the following table in the manuscript:
Table 1. Preliminary identification of derivatives extracted from CSw.
|
No. |
ID |
Formula |
Identification |
|
1 |
M197T422 |
C9H10O5 |
Syringic acid |
|
2 |
M359T569 |
C18H16O8 |
Rosmarinic acid |
|
3 |
M118T126 |
C15H24O |
Caryophyllene alpha-oxide |
|
4 |
M118T126 |
C5H11NO2 |
Betaine |
|
5 |
M439T818 |
C30H48O3 |
Oleanolic acid |
Lines 258-259 and 265-269 in the revised version.
6) Nuclear translocation images: The difference between cytosolic and intermediate classification is not clearly visible. Improving the quality of these images would be helpful for clearer interpretation.
Response: Thank you for your careful checks. We uploaded a clearer image instead of the original one. Line 228 in the revised version.
Reviewer 3 Report
Comments and Suggestions for Authors
The authors have significantly improved their manuscript. I accept the corrections made by the authors. I think that in this version the paper can be accepted for publication.
Author Response
We feel great thanks for your recognition of our work “Clerodendranthus spicatus (Thunb.) water extracts reduce lipid accumulation and oxidative stress in the Caenorhabditis elegans”.